# Fecal Metabolic Profiling of Breast Cancer Patients during Neoadjuvant Chemotherapy Reveals Potential Biomarkers

**DOI:** 10.3390/molecules26082266

**Published:** 2021-04-14

**Authors:** Oumaima Zidi, Nessrine Souai, Henda Raies, Farhat Ben Ayed, Amel Mezlini, Sonia Mezrioui, Fabrice Tranchida, Jean-Marc Sabatier, Amor Mosbah, Ameur Cherif, Laetitia Shintu, Soumaya Kouidhi

**Affiliations:** 1Department of Biology, Faculty of Sciences of Tunis, Farhat Hachad Universitary Campus, University of Tunis El Manar, Rommana, Tunis 1068, Tunisia; oumaima.zidi@hotmail.fr (O.Z.); nessrine.souai@fst.utm.tn (N.S.); 2Laboratory of Biotechnology and Valorisation of Bio-GeoRessources, Higher Institute of Biotechnology of Sidi Thabet, BiotechPole of Sidi Thabet, University of Manouba, Ariana 2020, Tunisia; amor.mosbah@gmail.com (A.M.); ameur.cherif@uma.tn (A.C.); 3Service d’Oncologie Médicale, Hôpital Salah-Azaïz, Tunis 1006, Tunisia; henda.rais@yahoo.fr (H.R.); soumayakouidhi15@yahoo.fr (A.M.); 4Association Tunisienne de Lutte Contre le Cancer (ATCC), Tunis 1938, Tunisia; farhatbenayed@gmail.com (F.B.A.); sonia.mezrioui@gmail.com (S.M.); 5Aix Marseille Univ, CNRS, Centrale Marseille, iSm2, 13284 Marseille, France; fabrice.tranchida@univ-amu.fr (F.T.); laetitia.shintu@univ-amu.fr (L.S.); 6Faculté de Pharmacie, Institute of NeuroPhysiopathology (INP), UMR 7051, 27, Boulevard Jean-Moulin, CEDEX, 13005 Marseille, France

**Keywords:** breast cancer (BC), metabolomics, gut microbiota, dysbiosis, metabolites, biomarkers

## Abstract

Breast cancer (BC) is the most common form of cancer among women worldwide. Despite the huge advancements in its treatment, the exact etiology of breast cancer still remains unresolved. There is an increasing interest in the role of the gut microbiome in modulating the anti-cancer therapeutic response. It seems that alteration of the microbiome-derived metabolome potentially promotes carcinogenesis. Taken together, metabolomics has arisen as a fascinating new omics field to screen promising metabolic biomarkers. In this study, fecal metabolite profiling was performed using NMR spectroscopy, to identify potential biomarker candidates that can predict response to neoadjuvant chemotherapy (NAC) for breast cancer. Metabolic profiles of feces from patients (*n* = 8) following chemotherapy treatment cycles were studied. Interestingly, amino acids were found to be upregulated, while lactate and fumaric acid were downregulated in patients under the second and third cycles compared with patients before treatment. Furthermore, short-chain fatty acids (SCFAs) were significantly differentiated between the studied groups. These results strongly suggest that chemotherapy treatment plays a key role in modulating the fecal metabolomic profile of BC patients. In conclusion, we demonstrate the feasibility of identifying specific fecal metabolic profiles reflecting biochemical changes that occur during the chemotherapy treatment. These data give an interesting insight that may complement and improve clinical tools for BC monitoring.

## 1. Introduction

Breast cancer (BC) is the most commonly occurring cancer and the second leading death cause in women worldwide. There are over 2.1 million new cases diagnosed annually all over the world [1]. Its incidence has risen to unprecedented levels in recent decades, making it the major public health problem of the world [2]. In Tunisia, it represents 33% of female cancers with 1600 new cases/year [3]. Despite the advances in the treatment of breast cancer, mortality from this disease is still high because current therapies (chemotherapy, radiology) are limited by the emergence of therapy resistance [4,5,6]. Breast cancer has particular epidemiological, diagnostic, and prognostic features. Its management is a veritable challenge taking into account major medical issues due to treatment protocols. Even if BC is treated early, women will always suffer from the side effects of anticancer treatments [7,8,9]. Thus, the identification of biomarkers for prognosis and treatment response may help stratify patients’ individualized treatment. Metabolomics has arisen as a fascinating new omics field to screen promising metabolic biomarkers.

Metabolomics is a state-of-the-art method with demonstrated effectiveness in numerous studies providing information about biological systems complementary to those provided by other “omics” approaches [10]. Metabolomics provides a very powerful tool for the discovery of clinically relevant biomarkers for the diagnosis, prognosis, and prediction of many diseases including BC. This approach will also allow the identification of the metabolites correlated to the modulation of responses to anticancer treatments, now referred to as pharmacometabolomics. The potential of pharmacometabolomics has already been demonstrated for different classes of anticancer drugs [11]. A few investigations have applied this approach to BC treatment. More recently, two pilot studies found that complete response to neoadjuvant chemotherapy in BC is correlated with decreased serum levels of methionine, glutamine, and linoleic acid [12,13]. In another study investigating the response to neoadjuvant chemotherapy (NAC) in HER2^+^ BC patients, patients with a pathological complete response showed high serum levels of spermidine and low serum levels of tryptophan compared with the poor responders [14]. In BC tumors, taurine, choline, and glycerophosphocholine were contributing metabolites for the prediction of pathological complete response to NAC [15,16]. At the urinary level, chemotherapy-sensitive patients showed a significant decrease of glycine, cysteine, histidine, cysteine, and tryptophan levels compared to chemotherapy-resistant patients [17]. Additionally, there is a paucity of studies that involve the use of metabolomic profiles of blood serum and urine [18,19,20] to discriminate healthy controls from BC subjects but also differentiate metastatic BC from early stages of BC [21,22]. Hence, all the previous investigations that thoroughly reported the application of metabolomics to BC have mainly explored urine and plasma samples but never fecal samples. One important concern that has emerged from recent findings is the contribution of the gut microbiome to the metabolome signature. Several previous metabolomic studies have proposed the use of gut microbiota metabolites in disease diagnosis, revealing key biomarkers in colorectal cancer [23,24], diabetes [25], and gastrointestinal disorders.

The microbiome could be one additional factor related to BC and has recently gained interest. A symbiotic relationship between host and microbiota is crucial to maintaining a balanced gut (eubiosis). This eubiosis confers benefits to the host in many key aspects of life by preserving host physiology and health. Indeed, gut microbiota exerts fundamental functions spanning from metabolic to immunomodulatory properties [26]. Disturbance of gut microbiota balance (dysbiosis), has been associated with different cancers including BC [27]. It is notable that dysbiosis not only contributes to cancer pathogenesis and progression but also influences the therapeutic outcome. There is growing evidence that gut microbiota and anti-cancer agents interact in a bidirectional fashion. This crosstalk impacts several key mechanisms such as translocation, immunomodulation, metabolism, enzymatic degradation, reduced diversity, and ecological variation [28]. The gut microbiota has been reported to affect the response to neoadjuvant chemotherapy (NAC) by modulating either efficacy or toxicity [29,30]. Chemotherapeutic agents like cyclophosphamide (CTX) exert their anti-neoplastic effects through a variety of immunological pathways. For instance, CTX and doxorubicin induce the translocation of selective gut commensal bacteria (*Enterococcus hirae, Lactobacillus johnsonii,* and *Barnesiella intestinihominis*) into secondary lymphoid organs, where they reduce immunosuppressive intra-tumoral T regulators and then improve the reduction of tumor growth due to the chemotherapy [28,31,32]. Furthermore, chemotherapeutic agents such as 5-fluorouracil and cyclophosphamide are toxic for the gut microbiota, causing its alteration either directly or by activating an immune response [33]. Anticancer treatments significantly affect the microbiota composition, therefore disrupting homeostasis and exacerbating the patient’s discomfort [34]. Another major inconvenience of anticancer treatments is the development of chemotherapy resistance, which is known as the first cause of chemotherapy’s failure against BC [35]. This imposes the urgency of finding an effective way to monitor the toxicity and increase the effectiveness of chemotherapy. However, the mechanisms underlying this interaction have not been fully investigated, particularly in BC.

Thus, in this preliminary pioneer study, we aimed to: (i) identify and characterize specific fecal metabolite profiles in BC patients following chemotherapy treatment and (ii) establish a noninvasive metabolomic approach in order to improve the monitoring of BC patients.

To our knowledge, this paper represents the first study of fecal metabolomic profile for breast cancer cases undergoing anticancer treatment.

## 2. Results

### 2.1. Patient Characteristics

All patients were recruited at the Institute of Salah-Azaïz, Tunisia. The fecal samples from eight patients before and during three cycles of chemotherapy were collected and analyzed by NMR. They were all diagnosed with an invasive ductal carcinoma (IDC) grade II and were estrogen- and progesterone-receptor-positive. It is worth noting that patients recruited in this study were selected to share as many factors as possible for statistical significance (same ethnicity, similar diet and age range, as well as an identical physical condition and treatment). The selected clinicopathologically homogenous group of patients when starting our study had an average age of 62.4. It was a representative group in the cancer center. Based on their response, our study population included six good-responders and two non-responders to neoadjuvant chemotherapy. The clinical characteristics of the study population are listed in Table 1.

### 2.2. Impact of Three Chemotherapy Cycles on the Differential Fecal Metabolites of Breast Cancer Patients

All the factors (diet, age, menopause state, chemotherapy impact, and BMI) were investigated, and only the treatment impact showed significant changes in the metabolomic profile between the different groups.

Untargeted metabolomics analysis based on 1H-NMR spectroscopy was applied to investigate fecal metabolite fluctuations in BC patients before and during three cycles of chemotherapy. After excluding missing values and data filtering, a total of 82 metabolites were identified in feces and used for subsequent statistical analysis (see Table 2).

Principal component analysis (PCA) was carried out to generate an overview of the variations between groups. Figure 1A shows the PC1 vs. PC2 score plot for all the samples. The PCA distribution did not exhibit any significant trend or difference between the groups. In order to determine the metabolites that contributed to the differences between C0, C2, and C3 groups, an orthogonal partial least squares discriminant analysis (OPLS-DA) was then performed on the NMR dataset. The 2D OPLS-DA score plots of fecal NMR profiles among the four groups showed that C0, C2, C3 groups could be distinguished clearly with good model fitness and predictability (R^2^Y = 0.993 and Q^2^ = 0.835, *p*-value = 6.35 × 10^−7^) (Figure 1B); however, the fecal profiles of the C1 and C0 groups overlapped partially with each other (Appendix A) indicating that drug treatment started to affect the patient fecal metabolome after the second cycle of treatment. Metabolite fluctuations between C0 and C1, reporting differences of the metabolites concentrations between the two groups and represented in the heatmap, showed the same behavior (Figure 3).

The robustness of our statistical model was tested using a Y-matrix permutation method for which 200 OPLSDA models with randomized Y-matrix were calculated. The permutation plot in Figure 2 shows that none of the random models led to better sensitivity and predictivity than the original model, hence validating the observed.

Biomarkers were selected between the C3 and C0 groups using variable importance in projection (VIP) values (>1.0) from OPLS-DA and false discovery rate (FDR; >0.5) (Table 3). As shown in Table 3 and according to the receiver operating characteristic (ROC) analysis, 27 differential metabolites in the feces were able to significantly distinguish the C0 group from C2 and C3 groups. In the fecal metabolic profile, acetate, butyrate, glycine, propionate, isovalerate, valine, glutamate, phenylacetate, aspartate, ethanol, threonine, valerate, creatinine, succinate, arabinose, and alanine were increased after chemotherapy. Conversely, lactate, fumaric acid, myo-inositol, ribose, and vanillate were significantly decreased (Figure 3).

The area under the receiver operating characteristic curve of the biomarkers, a valuable statistical tool that evaluates the sensitivity and the specificity of biomarkers to be used in disease diagnosis and prognosis, was also calculated (Table 3).

The significantly altered metabolites (FDR < 0.001) were used to analyze the differential metabolic pathways in the BC patients using MetaboAnalyst 4.0. The results revealed that the metabolic pathways of propanoate, glycolysis, amino acid, methane metabolism pyruvate, caffeine, tyrosine metabolism, lysine degradation, synthesis and degradation of ketone bodies, and beta-alanine metabolism were significantly altered (Figure 4).

As illustrated in Table 3, we found that the selected biomarkers of the fecal metabolic profile were influenced by the chemotherapy treatment even if its duration was short. Comparing the three studied groups, we noticed that we have a chemotherapy effect on the metabolic profile only from the second dose. Many metabolites were slightly decreased after treatment, and some others were remarkably increased in the C3 group compared to the C0 group. Furthermore, short-chain fatty acids (SCFAs) are specific products of the gut microbiota, and they are the metabolites that clearly underline little changes in the gut. In this study, the alterations of the SCFAs were more noticeable and tended to increase from the second treatment cycle (Figure 5).

### 2.3. Changes of Metabolite Levels in Breast Cancer Patients before and after Chemotherapy

In this study, all the patients completed three cycles of chemotherapy and underwent evaluation by Response Evaluation Criteria in Solid Tumors version 1.0 (RECIST 1.0) [36]. Six were good responders (GR) (partial response) to the anticancer treatment and two were poor responders (PR) (had progressive disease). To investigate the changes of metabolite profile between breast cancer patients before and after chemotherapy, a *t*-test was performed for each compound (Table 4).

The results showed that in good-responder patients, the levels of some amino acids (methionine *p* = 0.0148; valine *p* = 0.0120; alanine *p* = 0.0002; isoleucine *p* = 0.0201, and glutamate *p* = 0.0042), propionate (*p* = 0.0081), acetate (*p* = 0.0079), creatine (*p* = 0.0085), phenylacetate (*p* = 0.0236), 3-methylhistidine (*p* = 0.0063), histamine (*p* = 0.0105), ethanol (*p* = 0.0157), theophylline (*p* = 0.0379), and succinate (*p* = 0.0016) showed a significant increase after chemotherapy, which was not found in chemotherapy-insensitive patients. Moreover, hypoxanthine (*p* = 0.0087) showed significant decrease. After chemotherapy treatment in non-responder patients, uracil (*p* = 0.01349), tyrosine (*p* = 0.0059), and acetone (*p* = 0.0365) increased obviously, while the level of butyrate (*p* = 0.0165), methanol (*p* = 0.0343), and 3-methylhistidine (*p* = 0.0085) decreased significantly.

## 3. Discussion

Metabolomics is currently considered as a promising tool to explore the metabolic profile in BC, allowing for the potential identification of relevant biomarkers in chemotherapy management and monitoring. Gut bacteria can affect the response to chemotherapy by modulating either efficacy or toxicity [37]. Anticancer therapies themselves significantly affect the microbiota structure and function [37]. In this context, we conducted a comprehensive metabolic analysis in order to investigate the effect of adjuvant chemotherapy on the fecal metabolome and identify relevant biomarkers in BC patients.

According to our results, the fecal metabolome signature was significantly different in BC patients following two and three cycles of chemotherapy compared to BC during the first cycle or without treatment. Furthermore, altered levels of the 27 metabolites identified in this study reflect changes in the metabolic activity of several pathways and could be associated with drug effects. To the best of our knowledge, this is the first preliminary study in which NMR metabolomic analysis was performed to identify specific fecal metabolites in BC patients under neoadjuvant chemotherapy.

The most important changes in our findings were observed in amino acids (AAs) (glycine, valine, alanine, threonine, glutamate, aspartate, methionine, histidine, tryptophan, and aspartate) levels. These metabolites were downregulated in the pretreatment group (C0) but interestingly increased after chemotherapy administration (especially in C3 group). Amino acids have been shown to play an important role in the regulation of energy and protein homeostasis in the host body. Many researchers have demonstrated the diagnostic and the prognostic potential of amino acids in a range of human diseases such as diabetes and cancer [38,39]. Eniu et al. highlighted significant alterations of AA levels in the plasma of BC patients. They reported a quantitative decrease of five AAs (arginine, alanine, tryptophan, isoleucine, and tyrosine) and considered them as plasmatic biomarkers for BC detection [40]. In the urine samples, amino acids are downregulated in BC patients. Decreased levels of seven amino acids (alanine, isoleucine, threonine, cysteine, glutamic acid, tryptophan, and isoleucine) in the urinary profile of BC could be explained by the high demand of amino acids in tumor metabolism [41]. Alterations in the glycine pathway were highly correlated with the fast proliferation of breast cancer cells. Jain and his collaborators, by investigating metabolomic profile of BC tissues, demonstrated that glycine is highly consumed by rapidly proliferating cancer cells and released by slowly proliferating cells [42]. More importantly, recent work has identified that the glycine, serine, and threonine metabolism are central to cancer cell proliferation and breast cancer metastasis [43,44]. Glycine metabolism may, therefore, represent a vulnerable metabolite in breast cancer cells that could be targeted for therapeutic benefits. Recent research has correlated the alteration in the plasmatic profile of aspartate in BC to the consumption of this non-essential AA by the cancer cells. A study reported that aspartate levels were higher in cancer cells than in BC patients’ blood [45]. Few studies have investigated serum amino acid changes due to neoadjuvant chemotherapy in breast cancer. Wei et al. have identified a significant decrease in threonine, isoleucine, and glutamine in patients with good response compared to non-responders [12]. Miolo and coworkers have demonstrated that the good responders showed low amounts of plasmatic tryptophan compared with the poor responders [14]. Several studies have been performed on tumoral tissues to understand the metabolic pathways involved in breast cancer drug response, but few of them have underlined the implication of amino acids in chemotherapy response. For example, in a study conducted in triple-negative breast cancer cell lines, glutamine and glutamate increased whereas lysine, proline, and valine decreased in the presence of the anticancer agents [46].

In addition, we found that chemotherapy is correlated with decreased concentrations of lactate. It has been well established that lactate is a marker of tumor aggressiveness since high levels of lactate have been correlated to the high incidence of distant metastasis and low survival average [47,48]. In general, cancer patients are characterized by increased lactate production and high glucose consumption [49,50]. Altered energy metabolism, which is a biochemical fingerprint of cancer cells, has been suggested as one of the “hallmarks” of cancer. This metabolic anomaly was earlier described by Otto Warburg in the 1920s [51]. Warburg reported that tumor cells, compared to normal ones, converted glucose to lactate at high speed even in the presence of oxygen and thus maintained a high rate of glycolysis [52]. Moreover, in vitro studies showed increased glycolysis and lactate production is associated with chemoresistance in MCF-7 cells [53].

Succinic acid was upregulated while fumaric acid was downregulated in the feces of the C2 and C3 groups. This decrease has been reported in colon cancer and consistently related to the low levels of *Bacteroides* [54]. Both succinic and fumaric acids have been reported as oncometabolites or endogenous cancer-causing metabolites [55]. They are intermediate components of the Krebs cycle that significantly increases in cancer tissues [56]. Their accumulation in tumor cells supplies anabolic precursors for tumor growth and induces tumor aggressiveness by causing epigenetic changes such as the dysregulation in the anti-metastatic miRNA cluster mir-200ba429 [57]. The accumulation of succinate and fumarate were recently explained by the mutations in the genes coding succinate dehydrogenase (SDH) and fumarate hydratase (FH) enzymes of the TCA cycle and their inactivation [57,58].

Further, many studies assigned the chemotherapy response to the crosstalk between the gut microbiota and the anticancer agents [59]. Indeed, several studies found that neoadjuvant therapy efficacy is facilitated and abrogated by the gut microbiota. Cyclophosphamide (the most used molecule in the chemotherapeutic treatment of breast cancer) causes a shortening of intestinal villi and impermeability of the intestinal barrier, which facilitates the translocation of commensal bacteria such as *Enterococcus hirae* and *Barnesiella intestinihominis* to secondary lymphoid organs [60]. Once in the lymphoid organs, *E. hirae* helps to mediate CTX-driven accumulation of type 17 and type 1 T helper cell response, and *B. intestinihominis* increases systemic levels of a polyfunctional subset of cytotoxic CD_8_^+^ T cells [31,61]. Despite the crucial role of the gut microbiota in improving the chemotherapeutic response, anticancer agents can be toxic to the intestinal microbiome and cause its alteration. Recently, J.L. Alexander and coworkers found no reduction in the total bacterial counts in the gut after 7 days of chemotherapy, but they established a decrease in the abundance of lactobacilli and enterococci [28]. The gut microbiota directly metabolizes chemotherapeutic drugs and affects their pharmacokinetics, anticancer activity, and toxicity at various levels [62,63]. This has been demonstrated for several drugs, including 5-fluorouracil, epirubicin, and irinotecan [30,64,65]. Xenobiotics induce changes in the composition of the gut microbiota and further modulate its effect on drug metabolism. Biotransformation of drugs mediated by the gut microbiota includes many chemical reactions such as the nitroreduction of the radiation sensitizer misonidazole, the hydrolysis of the antimetabolite methotrexate, and the deconjugation of the liver-detoxified form of the topoisomerase I inhibitor irinotecan [66,67]. For example, irinotecan is activated by hydrolysis to form SN-38, an inhibitor of topoisomerase 1, which is later deactivated to SN–38–G in the liver by hepatic glucuronidation. SN–38–G is then excreted into the gut with bile. Within the gut lumen, bacterial β-glucuronidases reactivate it to its active enterotoxin form, which induces significant intestinal toxicity and diarrhea [68]. The β-glucuronidase activity was found mostly in *Clostridium* clusters XIV and IV [69]. The use of a specific antibacterial for bacterial β-glucuronidase has been shown to effectively treat intestinal inflammation induced by irinotecan therapy in experimental animals [65]. Gut microorganisms also have the potential to decrease the absorption of certain drugs by physical binding and segregation [70]. The fecal metabolome provides a functional readout of microbial activity and can be used as an intermediate phenotype mediating microbiome-drug interactions. As previously mentioned, short-chain fatty acids (SCFAs) are the end fermentation products of non-digestible carbohydrates by the gut microbiota. In the current study, SCFAs (propionate, acetate, and butyrate) tended to be upregulated in the stools of C2 and C3 groups vs. C0 group, showing that chemotherapy may play a key role in increasing the SCFAs producing bacteria. Recent studies mentioned a direct link between qualitative and quantitative changes of SCFAs and gut microbiota composition (alteration in the gut diversity) [71]. SCFAs are well-known biomarkers that promote apoptosis and inhibit invasive phenotypes in BC cells. Propionate, acetate, and butyrate are the three most predominant SCFAs. Among them, butyrate has been extensively investigated for its role in the suppression of colonic inflammation and carcinogenesis [72,73]. It is mainly metabolized for energy production in the colonic epithelium. Recent data suggest that it has functions at the level of gene expression, reducing cell proliferation and inducing differentiation and apoptosis [74]. Propionate and acetate also induce apoptosis, but less so than butyrate. Interestingly, butyrate increased the intracellular concentration of anticancer agents including 5-fluorouracil, doxorubicin, topotecan, and irinotecan, which improves their therapeutic efficacy and leads to cell apoptosis [75]. Recent data mentioned that acetate functions as a nutritional source for tumors and as a regulator of cancer cell stress. Thus, stopping its recapture by cancer cells may provide an opportunity for therapeutic intervention [76,77]. Previously, an increase of acetic acid concentrations has been demonstrated in breast cancer MDA-B-231 cells after chemotherapeutic treatment, suggesting that anticancer agents have a high influence on the recapturing of acetate by the BC cells [76]. The succinate pathway is the major route for propionate formation from dietary carbohydrates by Bacteroidetes. Propionate levels in feces were recently correlated with the relative abundance of Bacteroidetes [78].

In conclusion, it is worth noting that the current study was a qualitative and pilot study that aimed to establish a proof of concept of the application of the metabolic approach during chemotherapy treatment for breast cancer. Since this kind of study is relatively innovative, rather than concluding on the basis of statistical results, our objective was first to demonstrate the pertinence of this approach in terms of the investigation pathway to be considered in further studies and to determine the aspects to focus on. Despite the low number of samples, this study enabled us to find significant intraindividual differences among therapy cycles. This result by itself is promising since this qualitative study could identify key biomarkers and pathways that have been previously described to be involved in breast cancer and/or chemotherapy response. Further studies will be performed in the future to optimize more targeted metabolic analysis that focuses on specific metabolites or pathways for diagnostic and therapeutic implementation in clinical practice.

## 4. Materials and Methods

The study was reviewed and approved by the Ethics Committee of the Tunisian Association for the Fight against Cancer (Avis-01-2018 CE-ATCC). Written informed consent was obtained from each subject before participating in the study.

### 4.1. Sample Collection

This prospective work included 8 patients for whom a diagnosis of breast cancer was established with histological evidence and who will undergo the same chemotherapeutic treatment, FEC100 (5-fluorouracil; epiribucine; cyclophosphamide).

Fecal samples were self-collected by patients (following instructions provided by the study coordinator) just prior to treatment start and 20 days after every cycle to guarantee treatment impact on the metabolomic profile. Samples were collected in the early morning before the new treatment dose. All samples were stored at −80 °C until metabolite extraction.

### 4.2. Fecal Metabolite Extraction

Fecal water was extracted as described by Lamichhane and collaborators [79] with some modifications. Aliquots of about 125 mg thawed stool material were mixed with 1 mL phosphate-buffered saline (1.9 mM Na_2_HPO_4_, 8.1 mM NaH_2_PO_4_, 150 mM NaCl, pH 7.4) containing 90:10 D_2_O/H_2_O *(v/v*) for the field lock of the NMR spectrometer and 1mM of sodium 3-(trimethylsilyl) [2,2,3,3,-2H4] propionate (TSP) acting as a peak of reference at 0 ppm. The consistency of TSP signal integration was verified in order to validate the use of this signal as a reference for quantification. Mixtures were homogenized by vortexing for 1 min per sample. The homogenates were sonicated at ambient temperature (298 K) for 30 min to destroy bacterial cells. The fecal slurry was then centrifuged at 4 °C for 1 h at 18,000× *g*. Supernatants were collected and centrifuged at 4 °C for 15 min at 15,000× *g*. After centrifugation, 600 µL of supernatant was transferred into 5mm NMR tubes (if immediately used for NMR spectroscopy) or into labeled Eppendorf tubes and stored at −80 °C until analysis.

### 4.3. NMR Spectroscopy

NMR data were recorded using Bruker 600MHz AVANCE III Spectrometer equipped with a BBFO + probe and a Sample Jet autosampler, which enabled the storage of 5 racks of 96 NMR tubes at 5 °C.

The sample temperature was controlled at 300 K during experiments. Spectra were recorded using the 1D Nuclear Over Hauser Effect spectroscopy pulse sequence (trd-90°-t1-90°-tm-90°-taq) with a relaxation delay (trd) of 24 s, a mixing time (tm) of 4 ms, and a t1 of 4 μs. The sequence enables optimal suppression of the water signal that dominates the spectrum. We collected 128 free induction decays (FIDs) of 65,532 data points using a spectral width of 12,019.230 kHz and an acquisition time of 2.726 s. The spectra were automatically phased and baseline corrected and referenced to the internal standard (TSP; δ = 0.0 ppm).

The relaxation delay was set at 24 s in order to reach the complete relaxation of all the metabolites between scans; this is a mandatory step in NMR when absolute concentration of the metabolites is calculated.

Two-dimensional (2D) NMR spectra were obtained to aid the assignment of fecal metabolites. The set of 2D experiments included ^1^H-^1^H correlation spectroscopy (COSY); ^1^H-^1^H total correlation spectroscopy (TOCSY), and ^1^H-^13^C heteronuclear single quantum correlation (HSQC) using the standard parameters implemented in Topspin 3.5pl7 (Bruker Biospin GmbH, Karlsruhe, Germany).

### 4.4. NMR Data Processing

NMR data were further processed using NMRPROCFLOW v1.2 [80]. Basically, 1D NMR spectrum baselines were further adjusted using the global correction method. The spectral region with the residual water peak (4.5–5 ppm) was excluded from the data. To align the set of spectra, we chose the alignment method based on a least squares algorithm. The spectral region from 0.5 to 9 ppm was binned using an intelligent bucketing method with a resolution factor of 0.5. The resulting dataset was then normalized using the constant sum normalization method. Finally, normalized data were exported to SIMCA-P 14 (Umetrics, Umea, Sweden) prior to statistical analysis.

### 4.5. Metabolite Identification

Metabolites were identified based on their respective chemical shifts using a library provided with Chenomx NMR Profiler version 8.5 (Chenomx NMR Suite 8.5, Chenomx Inc., Edmonton, AB, Canada) and also using NMR databases (Madison Metabolomics Consortium Database) [81] and HMDB (Human Metabolomics Database) [82] and quantified using TSP signal as a reference. Following initial designation, metabolite identifications were confirmed conducting ^13^C and 2D NMR experiments (COSY and HSQC). The statistical analyses were performed on the matrix of the quantified metabolites.

### 4.6. Data Analysis

The normalized NMR dataset was unit variance scaled to highlight changes in low abundance metabolites. Initially, the principal component analysis (PCA) of the ^1^H NMR spectral data was carried out to identify any outliers within the dataset. Following PCA, orthogonal partial least squares discriminant analysis (OPLS-DA) was applied to optimize the separation between the different groups. The model robustness was evaluated through the calculation of R^2^Y (fraction of variance), Q^2^ (model predictability), and *p*-values. Close to 1, R^2^Y, and Q^2^ values indicate an excellent model, whereas low values are indicative of model over-fitting. The statistical model was tested for robustness with a Y-permutation performed using PLS-DA, which confirmed the observed metabolic variations. The statistical model was tested for robustness by a Y-permutation performed using PLS-DA, which confirmed the observed metabolic variations and by the use of a CV-ANOVA from SIMCA-P 16 (analysis of variance in the cross-validated residuals of a Y variable). The variable importance in projection (VIP) values of all peaks and the FDR from OPLS-DA were taken as coefficients for biomarker selection. The VIP value was higher than 1.0, so the variable was considered to contribute to the clustering of different groups in OPLS-DA. A hierarchical cluster analysis heat map was obtained using the ward clustering algorithm and Euclidean distance calculation to further confirm the results of PLS-DA and to show the distribution of metabolites among all individuals using MetaboAnalyst 4.0. NMR data were assessed for potential biomarkers in the first instance by constructing receiver operating characteristic (ROC) curves using MetaboAnalyst 4.0 for each metabolite. A ROC curve allows the simultaneous measurement of both sensitivity and specificity of every metabolite. ROC curves are often summarized into a single metric known as the “area under the curve” (AUC). For a perfect biomarker test, the AUC should be chosen as follows: 0.9–1.0: excellent biomarker; 0.8–0.9: very good; 0.7–0.8: good; 0.6–0.7: fair and <0.6: fail. The data obtained were subjected to an unpaired non-parametric test (Wilcoxon rank-sum test, also known as Mann–Whitney U-test) within MetaboAnalyst, and false discovery rates (FDRs), determined with significant analysis of microarray (SAM), which is essentially used for microarray data but also for metabolomic data (GC-MS; LC-MS and NMR compounds), were calculated to discover whether metabolites were significantly different between groups. As a final analysis step, a metabolomic pathway analysis (MetPA) was applied, using MetaboAnalyst 4.0, to all the metabolites to identify the most relevant pathways. The area of the circles is proportional to the effect of each pathway, with the color denoting the significance from the highest in red to the lowest in white. The difference between before and after chemotherapy was calculated with a paired-sample *t*-test (Excel 2019). A *p*-value below 0.05 was considered to be statistically significant.

## 5. Conclusions

Despite the low number of patients involved in this study, we showed here for the first time that NMR-based metabolomic analysis of fecal samples is a powerful method for the characterization of a neoadjuvant chemotherapy effect on BC patients. However, studies on larger patient cohorts with responding and non-responding patients are required to substantiate these findings. Furthermore, carrying out a metagenomic analysis would be necessary to better understand the impact of chemotherapy in BC patients and the implicated bacteria in the patients’ response to the treatment. The association of specific metabolic pathways with response prediction remains to be clearly understood. This preliminary study constitutes a promising first step metabolites, as AAs, lactate, fumaric acid, succinic acid, and SCFAs were clearly affected by the chemotherapeutic treatment and could represent potential predictors of drug response. Thus, the assessment of feces seems to be a promising non-invasive approach to reveal systematic metabolic variations and to contribute to more specific and sensitive insights related to anticancer treatment towards the use of NMR-based metabolomics and feces samples as a complementary tool for the prediction of breast cancer neoadjuvant chemotherapy response.

## Figures and Tables

**Figure 1 molecules-26-02266-f001:**
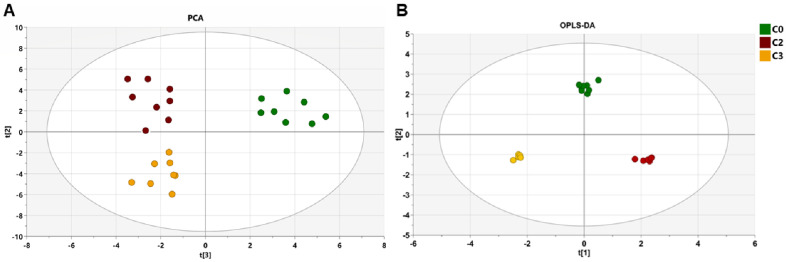
Principal component analysis (PCA) score plots and 2D orthogonal partial least squares discriminant analysis (OPLS-DA 2D) of the metabolic profiles from the fecal samples with a Pareto scale. (**A**): PCA for the three groups. (**B**) OPLS-DA 2D score plots of C0, C2, and C3 groups R2X = 0.656, R2Y = 0.993, Q2 = 0.835, and *p*-value = 6.35 × 10^−7^.

**Figure 2 molecules-26-02266-f002:**
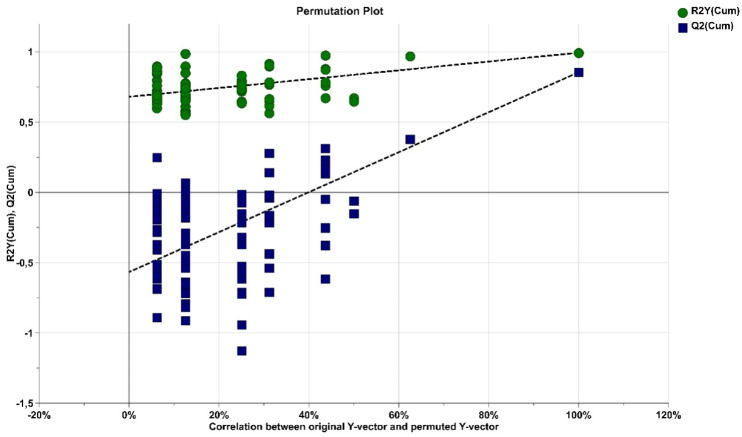
Permutation test plot for the OPLS-DA model (number of permutations, 200; Intercepts: R2 = 0.0, 0.669; Q2 = 0.0, −0.518).

**Figure 3 molecules-26-02266-f003:**
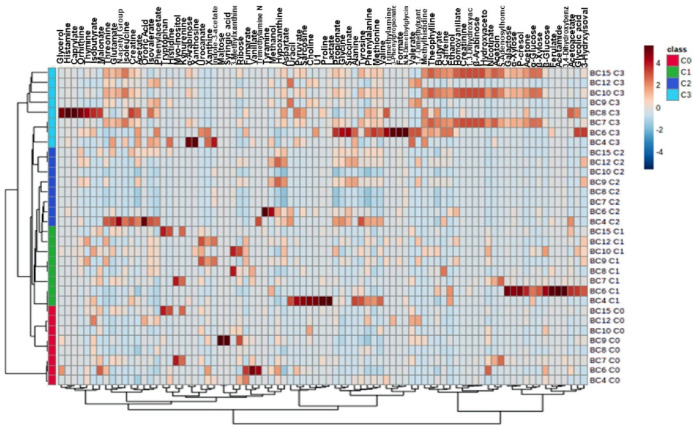
Heat map of differential metabolites among the groups. The color of each section represents the significance of the change of metabolites (red: upregulated; blue: downregulated). Rows: metabolites; columns: samples.

**Figure 4 molecules-26-02266-f004:**
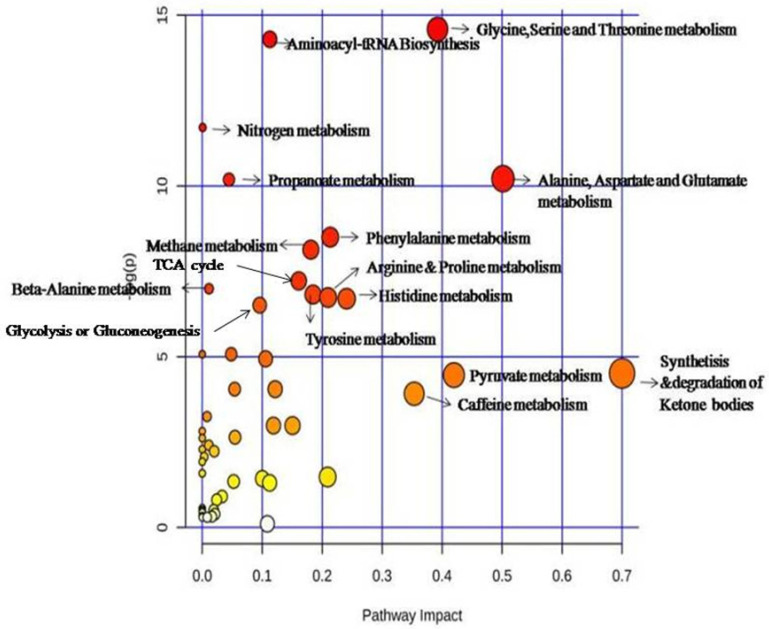
Metabolic pathway comparison between the different groups. Summary of the pathway analysis with MetPA with all the metabolites at each treatment cycle (C0, C2, and C3) were considered. The area with the bubbles is proportional to the effect of each pathway, with color denoting the significance from highest in red to lowest in white.

**Figure 5 molecules-26-02266-f005:**
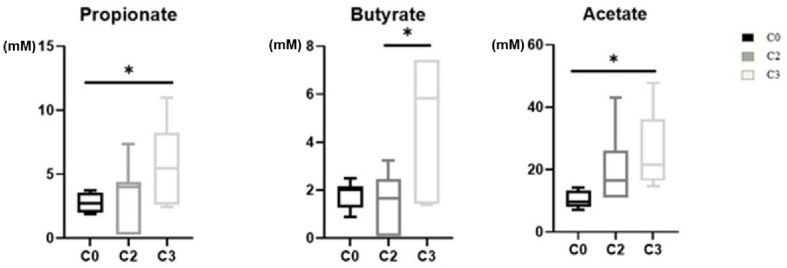
Metabolomic profile of SCFAs in the feces of BC patients undergoing NAC. After three cycles of NAC, three SCFAs (propionate, butyrate, and acetate) were significantly higher in the C3 group compared to the C0 group. SCFAs: short-chain fatty acids; NAC: neoadjuvant chemotherapy. *: indicates significant differences between the groups (*p*-value < 0.05).

**Table 1 molecules-26-02266-t001:** Clinical characteristics of the study population.

Variable	
Number	8
Sex (M/F)	F
Age	62.4 ± 4
BMI (Kg/m^2^)	28.04 ± 5
Infected Breast	2 L/5 R/1 both
Histology	IDC
Receptors	ER^+^, PR^+^
Grade	SBR II
Treatment	FEC
Treatment response	Six good-responders and two non-responders
Diet	Balanced diet
Physical activities	No physical activities
Collect time	Early morning

2 L: Left breast infected for two patients; 5 R: Right breast infected for five patients; 1 both: one patient with two infected breasts; IDC: invasive ductal carcinoma; FEC: 5-fluorouracil-epiribucine-cyclophosphamide; ER^+^: estrogen receptor^+^; PR+: progesterone receptor^+^.

**Table 2 molecules-26-02266-t002:** Assignments of detected metabolites ^1^H and ^13^C NMR.

Compound	^1^H Chemical Shift (ppm)	^13^C Chemical Shift (ppm)
Acetate	1.9204(s)	26.09
Acetoacetate	2.262 (s); 3.421 (s)	32.254; 56.1864
Acetone	2.299 (s)	33.4
Alanine	1.477 (d); 3.789 (q)	15.99; 21.86; 40.45
Alpha-Glucose	3.406; 3.531; 5.238 (d)	72.352; 74.166; 94.826
Alpha-Xylose	5.209 (d)	94.959
Arabinose	4.524 (d); 5.271 (d)	99.61; 96.14
Aspartate	2.684 (dd); 2.817(dd); 3.904 (dd)	39.23; 54.92; 177.04
Anserine	3.772 (s)	35.16
Beta-Glucose	3.249 (dd); 3.494; 4.652 (d)	76.854; 78.690; 98.643
Beta-Xylose	4.582 (d); 3.318	99.352; 76.753
Betaine	3.255 (s); 3.886 (s)	56.019; 68.839
Butyrate	0.898 (t); 1.561 (q); 2.16 (t)	15.99; 21.86; 40.45
Caffeine	3.353 (s); 3.511 (s); 3.947 (s)	30.609; 32.517; 35.967
Choline	3.199 (s)	56.618
Creatine	3.034 (s); 3.937 (s)	39.653; 56.535
Creatinine	3.046 (s); 4.06 (s)	32.936; 59
Desaminotyrosine	2.448 (t); 6.854 (d)	29.254; 115.4315
3.4-Dihydroxybenzeneacetate	3.382 (s); 6.692 (dd); 6.778 (d)	46.169; 124.12; 119.553
1.3-Dihydroxyacetone	4.413 (s)	67.535
1.7-Dimethylxanthine	3.305 (s); 3.926 (s)	30.375; 35.839
Ethanol	1.187 (t); 3.659 (q)	19.58; 60.167
Ferulic acid	6.899 (d)	117.8
Formate	8.46 (s)	
Fumarate	6.522 (s)	138.1
D-Galactose	4.084 (t); 5.262 (d)	74.591; 95.014
Glutamate	3.763 (m)	57.357
Guanosine	7.985 (s)	140.4
Glycerol	3.657 (m)	65.187
Glycine	3.564 (s)	44.308
Glycolate	3.930 (s)	63.929
Histamine	7.144 (s); 7.913(s)	119.02; 138.6
Histidine	7.085 (s); 7.852 (s)	119.6; 138.8
Homovanillate	3.439 (s); 3.85 (s); 6.756 (dd)	46.586; 58.594; 124.638
2-Hydroxyisovalerate	0.836 (d); 3.844 (d)	18.26; 79.98
3-Hydroxyisovalerate	2.354 (s)	2.354 (s)
Hypoxanthine	7.982 (s); 8.116 (s)	144.43; 140.08
Indole-3-acetate	3.645 (s); 7.617 (d)	36.548; 121.38
Isobutyrate	1.065 (d); 2.39 (m)	22.102; 39.672
Isoleucine	0.906 (t); 1.003 (d); 3.655 (d)	13.834; 17.411; 62.249
Isopropyl alcohol	1.177 (d); 4.024 (m)	26.325; 67.064
Isovalerate	0.904 (d); 2.058 (d)	24.67; 49.906
Kynurenine	3.707 (d)	41.678
Lactate	1.312 (d); 4.107 (q)	23.812; 72.127
Levulinic acid	2.39 (t)	34.012
Malonic acid	3.113 (s)	50.242
Maltose	5.21 (d); 5.387 (d)	94.613; 102.274
Methanol	3.36 (s)	51.571
Methionine	2.141 (s)	32.721
Methylamine	2.606 (s)	27.603
3-Methylhistidine	3.721 (s); 7.065 (s)	34.49; 127.022
Myo-Inositol	4.104 (t); 3.27	74.93; 77.16
N-Acetylglycine	2.029 (s)	22.322
N.N-Dimethylglycine	2.918(s); 3.709 (s)	46.229; 62.578
Nicotinate	8.942 (s); 8.253; 8.613	
Ornithine	3.034 (t)	42.2
Orotic acid	6.199 (s)	104.059
p-Cresol	2.247 (s)	22.0728
Phenylacetate	3.54(s); 7.304 (m); 7.374 (m)	47.174; 129.162; 131.868
Phenylalanine	7.336 (m); 7.374 (m); 7.434 (m)	132.09; 129.1; 131.8
Hydrocinnamic acid	2.495 (t); 2.898 (t)	41.873; 34.561
Propionate	1.059 (t); 2.184 (m)	12.872; 33.4
Pyruvate	2.366 (s)	29.21
Ribose	4.106; 4.217 (dd); 4.936 (d); 5.254 (d)	71.721; 73.411; 96.527; 99.032
S-Adenosylhomocysteine	8.373 (s)	139.752
Sarcosine	3.598 (s)	53.513
Succinate	2.423 (s)	36.915
Sucrose	5.39 (d)	94.899
Syringate	3.905 (s)	59.06
Theophylline	3.31 (s); 3.48 (s)	30.93; 32.95
Threonine	1.334 (d); 3.592 (d); 4.26 (m)	22.189; 63.272; 68.779
Trimethylamine N-oxide	3.249 (s)	62.259
Tyramine	2.933 (t); 6.908 (d); 7.223 (d)	34.654; 118.542; 133.054
Tyrosine	6.904 (d); 7.199 (d)	118.8; 133.3
Tryptophan	7.53 (d); 7.311 (m)	114.5; 128
U1 (Unknown)	1.813 (s)	25.922
Uracil	5.808 (d); 7.54 (d)	103.794; 146.5
Urocanate	6.397 (d); 7.273 (d); 7.786 (s)	124.4; 133.7; 140.4
Valerate	0.860 (t); 1.306 (m); 2.191 (m)	15.75; 24.548; 40.4
Valine	0.994 (d); 1.048(d); 3.616 (d)	19.365; 20.698; 63.082
Vanillate	3.898 (s)	55.895
Xanthine	7.947 (s)	141.4

S = singlet; d = doublet; dd = double doublet; q = quartet; m = multiplet.

**Table 3 molecules-26-02266-t003:** Biomarkers identified in fecal metabolic profiles of breast cancer (BC) patients. ^a^ Area under the receiver operating characteristic (ROC) curve of the biomarkers; ^b^ sensitivity and ^c^ specificity were calculated from the ROC curve.

Metabolites	VIP	C0-C2		C2-C3		ROC Analysis		
		FDR	FC	FDR	FC	^a^ AUC	^b^ Sensitivity	^c^ Specificity
Acetate	3.26	<0.001	3.9871	<0.001	4.049	1.0	1.0	1.0
Succinate	9.02	<0.001	0.003884	<0.001	0.30446	1.0	1.0	1.0
Lactate	2.90	<0.001	0.021075	<0.001	12.43	0.98438	0.938	1.0
Glycine	1.19	<0.001	0.14787	<0.001	3.5881	0.969	0.812	1.0
Ribose	2.16	<0.001	10.817	<0.001	0.015883	0.90625	0.695	1.0
Valerate	1.61	<0.001	16.569	<0.001	0.29447	0.89062	0.772	1.0
Alanine	2.55	<0.001	0.37297	<0.001	6.8116	0.89062	0.836	1.0
Valine	1.53	<0.001	120.41	<0.001	0.35782	0.875	0.699	1.0
Phenylacetate	1.63	<0.001	2.4325	<0.001	0.53583	0.85938	0.688	1.0
Ethanol	1.86	<0.001	0.37961	<0.001	2.1783	0.85938	0.671	1.0
Butyrate	2.39	<0.001	3.3069	<0.001	0.39693	0.84375	0.641	1.0
Isoleucine	2.64	<0.001	0.0039568	<0.001	0.22841	0.82031	0.547	1.0
Creatinine	1.99	<0.001	10.631	<0.001	0.16213	0.82031	0.751	1.0
Isobutyrate	3.15	<0.001	2259.1	<0.001	0.075646	0.8125	0.625	1.0
Arabinose	3.31	<0.001	87.253	<0.001	0.015883	0.75	0.762	0.936
Threonine	1.33	<0.001	2.1585	<0.001	2.2738	0.719	0.344	1.0
Xylose	2.83	<0.001	89.196	<0.001	1.9863	0.71875	0.638	0.938
Propionate	2.32	<0.001	4.7322	<0.001	0.6252	0.703	0.638	0.953
Glutamate	1.07	<0.001	0.0031898	<0.001	0.23818	0.6875	0.662	0.875
3.4-Dihydroxybenzeneacetate	1.35	<0.001	0.0099035	<0.001	838.3	0.6875	0.653	0.836
Isovalerate	1.78	<0.001	0.2328	<0.001	0.44677	0.63281	0.632	0.895
Aspartate	1.59	<0.001	0.089599	<0.001	0.5052	0.625	0.65	0.812
Ornithine	1.48	<0.001	0.17305	<0.001	83.958	0.625	0.536	0.844
Methanol	2.86	<0.001	0.1776	<0.001	3.8184	0.61719	0.344	0.863
Fumarate	1.73	<0.001	12.937	<0.001	3.2866	0.60938	0.635	0.961
N.N-Dimethylglycine	1.77	<0.001	0.032145	<0.001	7.3135	0.58594	0.634	0.875
Theophylline	1.46	<0.001	0.41299	<0.001	0.09096	0.52344	0.625	0.766

FDR: false discovery rate; FC: fold change; VIP: variable importance in the projection; AUC: area under the curve.

**Table 4 molecules-26-02266-t004:** Changes of the metabolite levels before and after the chemotherapy.

Compounds	GR *p*-Value *t*-Test after vs. before Chemotherapy	PR *p*-Value *t*-Test after vs. before Chemotherapy	Compounds	GR *p*-Value *t*-Test after vs. before Chemotherapy	PR *p*-Value *t*-Test after vs. before Chemotherapy
Alanine	0.002 ↑	0.0869	Lactate	0.1021	0.1166
Succinate	0.0016 ↑	0.1734	1.3-Dihydroxyacetone	0.1033	0.2504
Glutamate	0.0042 ↑	1	Arabinose	0.1049	0.25
Tyrosine	0.1612	0.0059 ↑	Betaine	0.1266	0.25
Fumarate	0.0061 ↓	0.1907	Valerate	0.1274	0.25
3-Methylhistidine	0.0063 ↑	0.0085 ↓	Alpha-Xylose	0.1319	0.2102
Acetate	0.0079 ↑	0.1554	Vanillate	0.1328	1
Propionate	0.0081 ↑	0.1063	1.7-Dimethylxanthine	0.1415	1
Creatine	0.0085 ↑	0.1248	D-Galactose	0.1513	0.25
Hypoxanthine	0.0087 ↓	0.2471	Methylamine	0.1514	0.25
Histamine	0.0105 ↑	0.2404	Beta-Xylose	0.1554	0.1920
Valine	0.0120 ↑	0.2015	Beta-Glucose	0.1570	0.2840
Methionine	0.0148 ↑	0.0506	Formate	0.1587	0.2533
Ethanol	0.0157 ↑	0.25	Isobutyrate	0.1623	0.2534
Glycine	0.0160 ↑	0.2299	Alpha-Glucose	0.1663	0.2796
Butyrate	0.0514	0.0165 ↓	N.N-Dimethylglycine	0.1663	0.2601
Isoleucine	0.0201 ↑	0.1476	Levulinic acid	0.1746	0.7987
Phenylacetate	0.0236 ↑	0.2791	Choline	0.17854	0.2934
U1 (Unknown)	0.0322	0.25	Anserine	0.1816	0.25
Methanol	0.1542	0.0343 ↓	Isopropanol	0.1816	0.25
Uracil	0.0885	0.0348 ↑	Tyramine	0.181609	1
Acetone	0.0831	0.0365 ↑	Trimethylamine N-oxide	0.1816	1
Theophylline	0.0379 ↑	0.1693	Orotic acid	0.1816	1
Isovalerate	0.0560	0.3306	Maltose	0.1816	1
Ribose	0.0602	0.4583	3-Hydroxyisovalerate	0.2066	0.25
Caffeine	0.0619	0.2692	Glycolate	0.2383	0.25
Sarcosine	0.0651	0.0832	Desaminotyrosine	0.2466	0.25
Homovanillate	0.0667	0.25	Myo-Inositol	0.2704	1
Sucrose	0.0778	0.7901	Ferulic acid	0.2793	1
Pyruvate	0.2468	0.0847	Xanthine	0.2866	0.2113
Creatinine	0.0864	0.1824	Tryptophan	0.2849	0.2044
Aspartate	0.0873	1	Acetoacetate	0.3167	0.1917
N-Acetylglycine	0.0873	0.25	Urocanate	0.3255	0.1091
p-Cresol	0.0873	0.2596	S-Adenosylhomocysteine	0.3265	0.25
Lysine	0.0873	1	Ornithine	0.3321	0.2592
Malonic acid	0.0882	0.2803	3,4-Dihydroxybenzeneacetate	0.3632	1
Phenylalanine	0.3103	0.0964	Syringate	0.3837	0.3645
Glycerol	0.0991	0.2764	Nicotinate	0.4037	0.3360
Threonine	0.1318	0.0990	Histidine	0.4625	0.25
Proline	0.0996092	0.25	Guanosine	0.5633	0.1791

↓ Significant decrease in breast cancer patients after chemotherapy compared with before chemotherapy (*p* < 0.05); ↑ Significant increase in breast cancer patients after chemotherapy compared with before chemotherapy (*p* < 0.05).

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
