# Peer review of "Fecal Metabolic Profiling of Breast Cancer Patients during Neoadjuvant Chemotherapy Reveals Potential Biomarkers"

_molecules, 2021, doi:10.3390/molecules26082266_

Round 1

Reviewer 1 Report

Manuscript entitled 'Fecal metabolic profiling of breast cancer patients during neoadjuvant chemotherapy reveals potential biomarkers' by Zidi et al. is an interesting study and the authors presented their data quite well in this present format of this manuscript. But there are some issues which needed to be addressed before this manuscript can be accepted in "Molecules''

  1. Authors took samples from only 8 patients, which is too low in my understanding. Kindly justify.
  2. Samples from the patients has been taken prior treatment and after 20 days of every cycle. Any specific reason for collecting samples after 20 days of each cycle?
  3. The average age group of the petients is 62.4. Why did authors chose this particular age group?
  4. Fecal metabolites profiling is also depending on the individual gut microbiome (depends on their diet, age, physical condition). So the study of gut microbiome of individual is important to monitor the metabolites released through fecal. Authors should address this point in their manuscript.
  5. Did the authors try to look for any kind of secondary metabolites excreted through fecal materials of the patients before and after treatment? Including secondary metabolites profile will make this study more valuable. 

Reviewer 2 Report

The manuscript titled “Fecal metabolic profiling of breast cancer patients during neoadjuvant chemotherapy reveals potential biomarkers” describes the study of fecal metabolite profiling using NMR Spectroscopy, to identify potential biomarker that can predict response to neoadjuvant chemotherapy for breast cancer. Authors claim that they identify specific fecal metabolic profiles that reflects biochemical changes upon chemotherapy treatment in patients with breast cancer.

The manuscript is well-written and describes the procedure and the metabolomic results.

In fact, authors identify specific fecal metabolic profiles that reflects biochemical changes upon chemotherapy treatment in patients with breast cancer. However there has to be some caution when trying to identify biomarkers.

In my opinion there are 3 major points of revision:

The major problem that I found with the dataset is that the patient number (n = 8; 6 good-responders and two non-responders to neoadjuvant chemotherapy) that should be increased in order to have a clear statistical correlation. Conclusions taken in biomarkers identification in fecal metabolic profiles of Breast cancer patients should be more cautious.

C0 samples corresponds to fecal sample before chemotherapy sessions but these patients have already breast cancer. Can the C0 fecal metabolome be related with the cancer itself? The control samples could be selected among female patients with no breast cancer diagnosis?

With such a reduced number of samples a supervised multivariate analysis tool is not the better choice. In fact, the is no discrimination with PCA. OPLS-DA is a supervised method that can bias the results showing a good discrimination that is not true.

Reviewer 3 Report

Zidi et al., performed NMR-based metabolomics of feces samples from breach cancer patients during chemotherapy. They compared fecal metabolite profiles between before chemo, and after 1, 2, 3 cycles of chemo, and found several metabolites increased or decreased after multiple chemo cycles. The findings are interesting, manuscript is well-written, and statistical analysis is appropriate. Below are concerns and comments.

1) N=8 is a very small number for a metabolomics study. In particular, N=2 for non-responders is not high enough for statistical analysis. The reviewer understand this is a pilot study, but if the authors can increase N, the study would be more meaningful.

2) Fecal metabolites are highly affected by diet, fasting/feeding cycle, and circadian rhythm. Can the authors provide patients' diet information? What time of the day the samples were collected? 

3) Fig.3 heatmap showing the average of all patients is not very informative to understand data variability, which is critical for human studies. Please provide heatmap for individial patients for these metabolites. Also, the image resolution is poor (hard to read metabolite names). Please correct.

4) In the text, the authors explained that C0 and C1 groups show similar metabolite profiles. This is very interesting and suggests that it takes time for chemodrugs to alter fecal metabolome. Given that this is one of the major findings of this study, please provide the data.

5) Fig.5. What is y-axis? concentrations (mM)?

6) Alphabetical ordering of metabolites is a good idea for Table 2. However, for Table 3 and 4, which shows significant metabolites, it is more appropriate to show the metabolites with the highest significance first and the next, and so on. 

Round 2

Reviewer 1 Report

The revised manuscript has addressed the review comments, which helps improve the quality of this work. Hence this revised manuscript is recommended for publication in its current format.

Reviewer 2 Report

Authors clarified all my comment.